# Evaluation of sample pooling for screening of SARS CoV-2

**Andargachew Mulu, Dawit Hailu Alemayehu, Fekadu Alemu, Dessalegn Abeje Tefera, Sinknesh Wolde, Gebeyehu Aseffa, Tamrayehu Seyoum, Meseret Habtamu, Alemseged Abdissa, Abebe Genetu Bayih, Getachew Tesfaye Beyene** *

Armauer Hansen Research Institute, Addis Ababa, Ethiopia

* getachew.tesfaye@ahri.gov.et

## Abstract

### Background

The coronavirus disease 2019 (COVID-19) pandemic has revealed the global public health importance of robust diagnostic testing. To overcome the challenge of nucleic acid (NA) extraction and testing kit availability, an efficient method is urgently needed.

### Objectives

To establish an efficient, time and resource-saving and cost-effective methods, and to propose an *ad hoc* pooling approach for mass screening of SARS-CoV-2.

### Methods

We evaluated pooling approach on both direct clinical and NA samples. The standard reverse transcriptase polymerase chain reaction (RT-PCR) test of the SARS CoV-2 was employed targeting the nucleocapsid (N) and open reading frame (ORF1ab) genomic region of the virus. The experimental pools were created using SARS CoV-2 positive clinical samples and extracted RNA spiked with up to 9 negative samples. For the direct clinical samples viral NA was extracted from each pool to a final extraction volume of 200μL, and subsequently both samples tested using the SARS CoV-2 RT-PCR assay.

### Results

We found that a single positive sample can be amplified and detected in pools of up to 7 samples depending on the cycle threshold (Ct) value of the original sample, corresponding to high, and low SARS CoV-2 viral copies per reaction. However, to minimize false negativity of the assay with pooling strategies and with unknown false negativity rate of the assay under validation, we recommend pooling of 4/5 in 1 using the standard protocols of the assay, reagents and equipment. The predictive algorithm indicated a pooling ratio of 5 in 1 was expected to retain accuracy of the test irrespective of the Ct value samples spiked, and result in a 137% increase in testing efficiency.

**Data Availability Statement:** All relevant data are within the manuscript and Supporting Information files.

**Funding:** The author(s) received no specific funding for this work.

                1 / 11

**Competing interests:** The authors have declared that no competing interests exist.

## Conclusions

The approaches showed its concept in easily customized and resource-saving manner and would allow expanding of current screening capacities and enable the expansion of detection in the community. We recommend clinical sample pooling of 4 or 5 in 1. However, we don't advise pooling of clinical samples when disease prevalence is greater than 7%; particularly when sample size is large.

## Background

The coronavirus disease 2019 (COVID-19) pandemic has revealed the global public health importance of efficient diagnostic testing [1, 2] to differentiate severe acute respiratory syndrome coronavirus 2 (SARS CoV-2) from other routine respiratory infections and to guide appropriate public health and individual clinical management [1]. Detecting carriers of the virus at various population levels is fundamental to response efforts. It ensures the quarantine of COVID-19 patients to prevent local community transmission, and more broadly informs national response team to take measures [3]. However, it remains uncertain whether there may have been community circulation of SARS CoV-2 prior to the identification of individuals with positive results through standard public health surveillance as detection and monitoring capacity is limited [4]. Testing in Ethiopia is generally done on handful of facilities while potentially infectious carriers at the community remain undiagnosed. Given the limited testing capacity available in Ethiopia, the decision to test is based on clinical and epidemiological factors and linked to an assessment of the likelihood of infection. However, testing of appropriate specimens from patients meeting the suspected case definition for COVID-19 is a priority for clinical management and outbreak control [4]. Thus, it is necessary to come up with new ways to efficiently and effectively use available resources.

Sample pooling (mixing of samples and testing at a single pool, but subsequent testing of individual samples is needed only if the pool tests positive) has been used as an attractive method for community monitoring of infectious diseases as it requires no additional training, equipment, or materials [5–7]. The key principles for successful application of group testing involve knowledge of the limit-of-detection, sensitivity, and specificity of the assay, and the prevalence of disease in a given population [8, 9]. Here we have shown a proof-of-concept for direct clinical sample and NA pooling for the diagnosis of SARS CoV-2 in Ethiopia using the existing assay.

## Objective

To establish an efficient, time and resource-saving and cost- effective methods and to propose an *ad hoc* laboratory-based surveillance approach for mass screening of SARS-CoV-2

## Materials and methods

### Design

The workflow comprises pooling of clinical respiratory samples and NA extraction, and extraction of NA from individual respiratory samples (Nasopharyngeal or oropharyngeal swabs in viral transport medium), followed by pooling of individually extracted NA samples. Then, conduct SARS-CoV-2 specific real-time RT-PCR using the Novel Coronavirus 2019-nCov PCR Kit-fluorescent PCR method of Da An Gene Co., Ltd, China, which is used

currently for the diagnosis of SARS CoV-2 in the country. Nucleic acid was extracted from 200 μL respiratory specimen using the NA extraction and Purification Reagent, DAAN Gene Co., Ltd, as recommended by the manufacturer (Da An Gene Co., Ltd, of Sun Yat-Sen University, China). All laboratory procedures (Sample processing: NA extraction and purification, master mix (MM) preparation, mixing of NA and MM, amplification/detection and analysis) were performed according to the manual provided by the manufacturer (Da An Gene Co., Ltd). Throughout the experiment we used BioRad CFX96 Deep Well Real-Time System, BioRad Laboratories, Inc, Singapore and program. Change in cycle threshold (ct) value (which is defined as the ct value of a reaction when the fluorescence of a PCR product can be detected above the background signal) of positive sample were analyzed. A Ct value is inversely proportional to the amounts of viral RNA in a reaction. The assay targets N and ORF 1ab region of SARS CoV-2. With this assay, a positive SARS CoV-2 result is determined when both targets reach a Ct value of ≤40, along with a Ct value of ≤32 and 40 for positive control and internal control, respectively.

## Pooling

We conducted the pooling in two arms (direct clinical samples arm and nucleic acid arm), and experiments for direct clinical samples were done in triplicate. The total number pools done on direct clinical sample were 54 using two positive and 18 negative samples. For the nucleic acid arm, a total of 18 pools were conducted using two positive and 16 negative samples.

First, we pooled direct clinical samples of previously known positive samples with low and high ct values up to 10 samples in 1 prior to NA extraction step (maximum dilution factor of 10), to a final extraction volume of 200μL when combined with an increasing number of confirmed negative samples (Table 1).

In this study, a positive sample with ct value ≤ 32 considered as low ct value, between [> 32 and ≤ 34] medium, and between [>34 and ≤40] is high. Then, NA was extracted from final pooled samples of 200μL with a final elution volume of 50 μL. From the eluate template NA, 5 μl was mixed with 20 μl of the RT-qPCR reagent master mix to have a final volume of 25 μl reaction mixture. Then, change in ct value of the positive control, positive samples, and the cycle when all tested with no ct value were analyzed.

Second, we pooled individual NA preparation extracted earlier from 200μL of direct clinical samples. To minimize pipetting error during the pool assembly, we took the same high volume (3μl) of extracted RNA as indicated in Table 2.

**Table 1. Direct clinical sample pools tested for SARS-CoV-2 RNA.**

| Pooling proportion (Dilution) | Volume of positive sample (μl) | The sum volume of negative samples in a pool (μl) |
|---|---|---|
| 1 (Original) | 200 | 0 |
| 1:1 (2 in 1) | 100 | 100 |
| 1:2 (3 in 1) | 67 | 133 |
| 1:3 (4 in 1) | 50 | 150 |
| 1:4 (5 in 1) | 40 | 160 |
| 1:5 (6 in 1) | 34 | 166 |
| 1:6 (7 in 1) | 29 | 171 |
| 1:7 (8 in 1) | 25 | 175 |
| 1:8 (9 in 1) | 23 | 177 |
| 1:9 (10 in 1) | 20 | 180 |

**Table 2. Nucleic Acid (NA) pooling design: Known positive individually extracted RNA and known negative samples separately extracted NA were pooled and a final volume of 5μl was taken from each pool to the master mix for amplification and detection.**

| Pooling proportion | Volume of positive RNA (μl) | Sum volume of known negative NA (μl) |
|---|---|---|
| original | 5 | 0 |
| 1:1 (2 in 1) | 3 | 3 |
| 1:2 (3 in 1) | 3 | 6 |
| 1:3 (4 in 1) | 3 | 9 |
| 1:4 (5 in 1) | 3 | 12 |
| 1:5(6 in 1) | 3 | 15 |
| 1:6 (7 in 1) | 3 | 18 |
| 1:7 (8 in 1) | 3 | 21 |
| 1:8 (9 in 1) | 3 | 24 |
| 1:9 (10 in 1) | 3 | 27 |

For detecting a single positive sample within a pool of negative nucleic acid extracts, we evaluated the ability of the standard qRT-PCR test under the protocol recommended by manufacture of the kits. Then, change in ct value of samples with low and high ct value was analyzed.

To assess the pool testing strategy, the most optimal testing configuration pool size was calculated using a Shiny App for pooled testing of Hierarchical algorithm (https://www.chrisbilder.com/shiny). As per the key principles of pooling, the following assumptions with numeric parameters were taken in to consideration: an experimental prevalence rate of SARS CoV-2 in Ethiopia to be 0.05 (whereas the observed positive rate within the tested individuals is reaching to 0.66% in the last 5 weeks), a two-stage pooling in a range of pool sizes 3–10 samples, an assay limit of detection (LOD) of 2.5 RNA copies/μL of reaction, an assay sensitivity of 98% -100% and an assay specificity of 100%. With these calculations, a pool size of 5 samples predicted and would provide the largest reduction in the expected number of tests of 58% when compared to testing clinical samples separately (Table 3).

To ensure the quality of the work during pooling, one staff member had been overseeing the pool assembly process, and had mitigated potential laboratory errors. Furthermore, experiments for direct clinical samples were done in triplicate. To avoid potential pipetting errors,

**Table 3. A comparison of the influence of optimal pool size on test efficiency* when the disease prevalence rate is 0.05.**

| Optimal sample pool size | Expected number of tests reduced (%) | Expected increase in testing efficiency (%) |
|---|---|---|
| 3 | 53 | 111 |
| 4 | 57 | 132 |
| 5 | 58 | 137 |
| 6 | 57 | 135 |
| 7 | 56 | 128 |
| 8 | 55 | 120 |
| 9 | 53 | 111 |
| 10 | 51 | 103 |

*Calculated using Shiny application of pooling strategy available at http://www.chrisbilder.com/shiny with the specified key principles of pooling indicated above. Expected increase in test efficiency is obtained by dividing expected number of tests reduced by expected number of tests per individual.

we used relatively higher volume of dilutions for RNA pooling. Moreover, our laboratory is participating in an external quality assessment program and have the approval that results we produce are reliable.

## Data analysis

To check if the variation between and within our experiments is statistically different, we run one-way Analysis of Variance (ANOVA) using an excel add-in program known as Analysis ToolPak.

## Ethical approval

The study is approved by the Armauer Hansen Research Institute/ALERT Ethics Review Committee.

## Results

With our pooling strategy, we were able to detect SARS CoV-2 positives samples in pooling up to 8 in 1 which tested positive in individual RT-PCR (Figs 1 and 2).

The results showed that pooled samples were positive within a range of 0 Ct to 6.75 Ct value difference from the original samples. Briefly, a total of 54 pools on direct clinical specimens each containing one positive sample were group tested. Of these pools conducted with positive samples with originally low ct value (high viral copy number), their Ct values were within a range of 29.28 to 35.67 for nucleocapsid (N) gene (Fig 1A and S1 Table) and 29.61 to 38.88 for the open reading frame (ORF)1ab genes (Fig 1B and S1 Table), where the highest dilution is 10 samples in 1 pool.

Similarly, the pools ct value for the SARS CoV-2 positive samples with originally high ct value (low viral copy number) were within a range of 35.09 to 38.67 for N gene and 37.43 to 40.00 for ORF1ab (S2 Table).

Overall, the average variance between the experiments and within the experiments is not statistically different. For the pooling experiments done with low Ct values of positive samples, the average variation of Ct values of N gene between the experiments is 0.032 while within the experiment is 2.258. For ORF1ab gene the average variation between experiments and within experiments is 0.67 and 3.05, respectively. Likewise, for pools done with high Ct values samples, the average variation for N gene between experiments is 0.02 and the variation within experiments is 1.92. and for ORF1ab gene, Ct value average variation is 0.16 and 0.75 in their order between and within experiments.

In our RNA pool, we were able to detect SARS-CoV-2 positives samples in pooling of up to 10 in 1(Fig 2A and 2B). Nine pools were done using SARS CoV-2 positive samples with original low ct value, and the Ct value of the pools range from 29.27 to 30.93 for N gene and from 30.41 to 32.63 for ORF1ab gene. Strictly speaking, the results show that RNA pooled samples were positive within a range of 1.53 to 3.19 and 1.23 to 3.45 Ct value difference from the original samples for N and ORF1ab genes, respectively (S3 Table).

In addition, the RNA pool experiments conducted with a positive sample of original high ct value, the pools Ct value ranges from 34.39 to 39.23 for N gene and from 36.58 to 38.86 for ORF1ab in 10 in 1 pool (Fig 3A and 3B). Similarly, the results show that RNA pooled samples were positive within a range of 0.48 to 4.48 and 0.95 to 3.20 Ct value differences from the original sample for N and ORF1ab genes, respectively (S4 Table).

As clearly seen in the figures above, as the number of negative pooled samples increases, the amplified RNA reaches the threshold later, which is expected from a diluted sample with the principle of sample dilution effect. However, in pools that were conducted using a positive

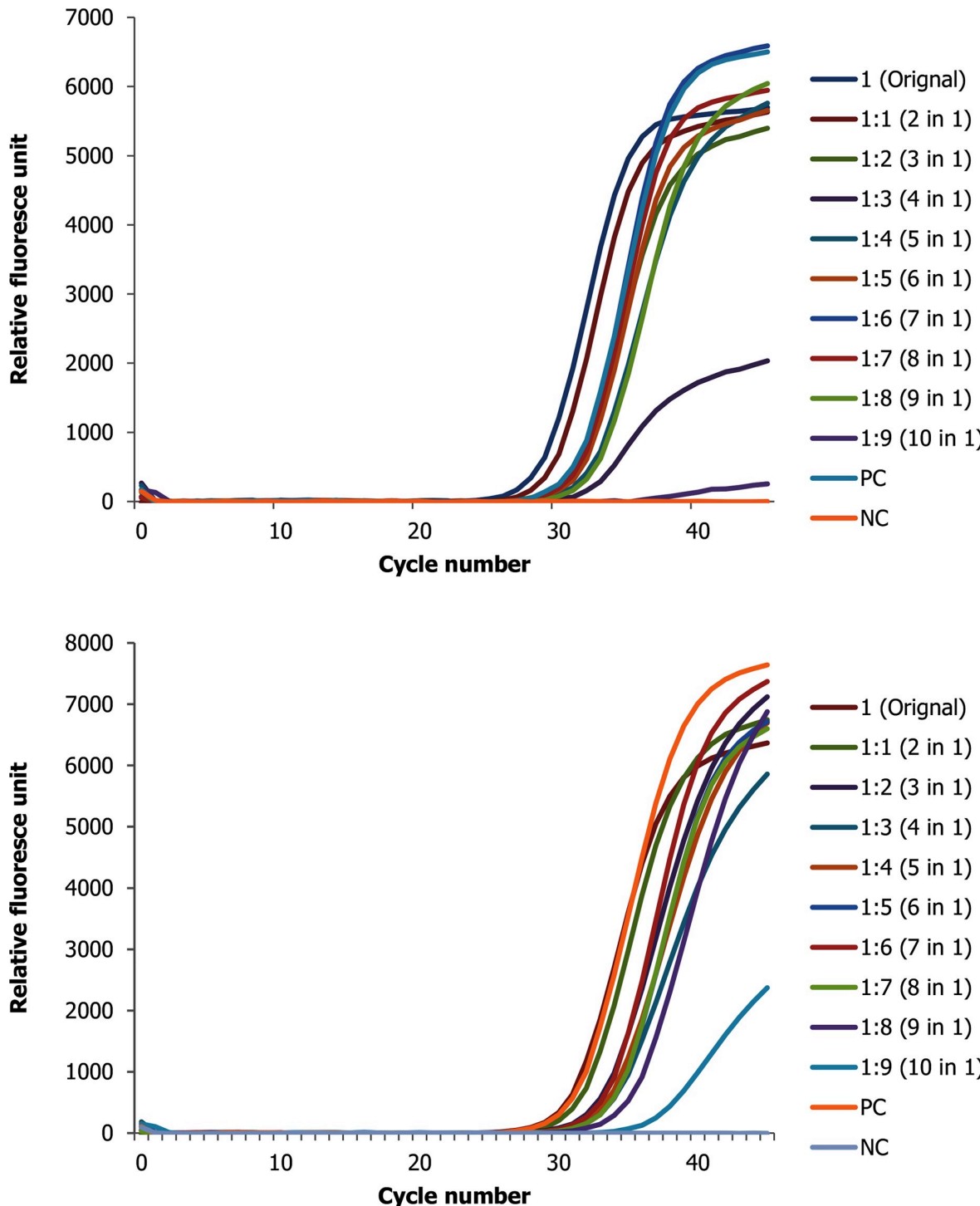

**Fig 1.** Change in ct value of positive direct biological sample with low ct value (high viral copy) spiked with up to 9 negative samples for the two target genes (A N and B ORF 1ab genes).

sample with low viral copy number the result for ORF1ab gene tends to be negative at higher level of dilution. For instance—out of the three replicate experiments, two of them revealed no Ct value or negative test result. Furthermore, when a positive sample spiked with nine negative

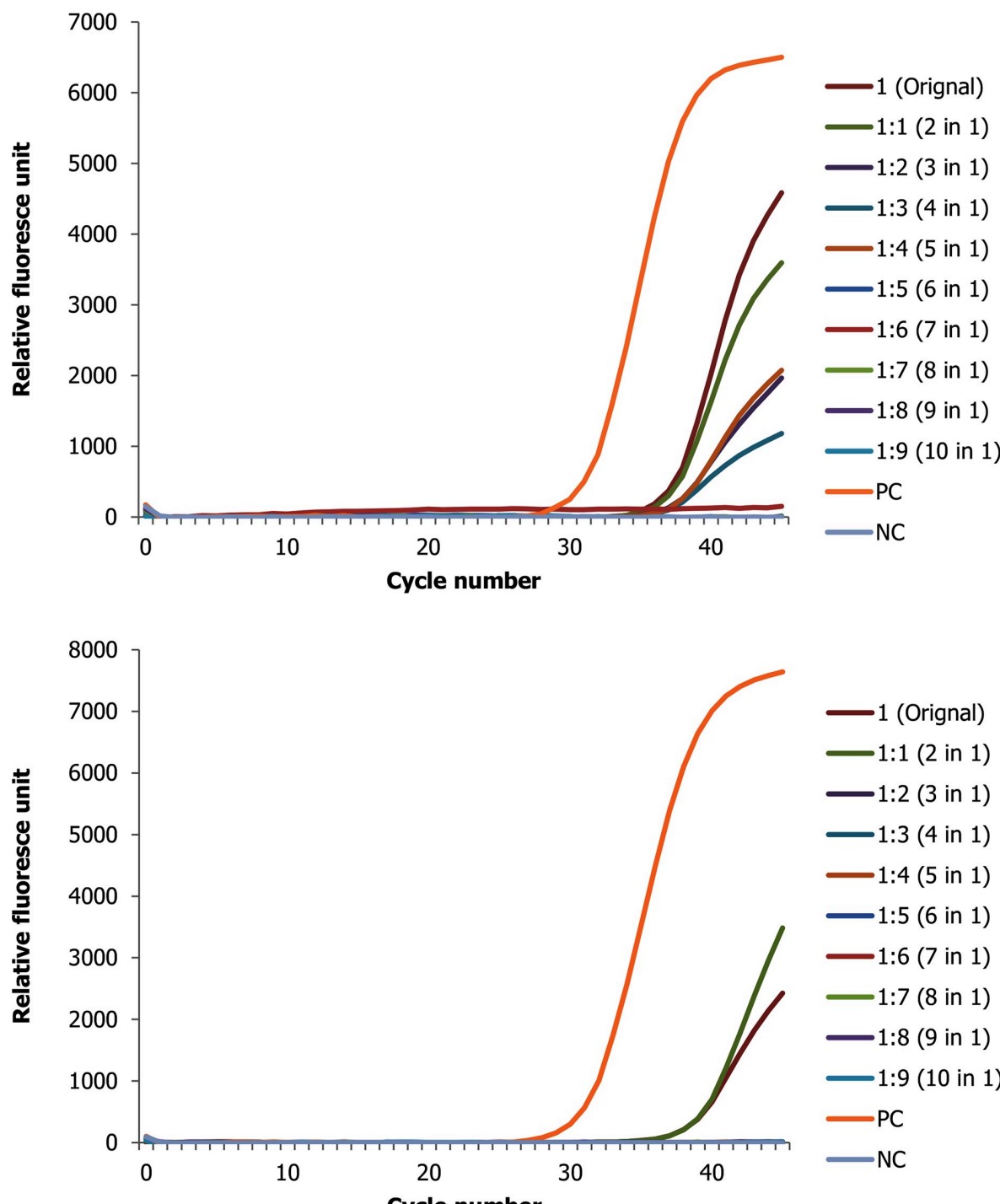

**Fig 2.** Change in ct value of RNA positive sample with low ct value (high viral copy) spiked with up to 9 negative samples for the two target genes (A N gene and B ORF1ab gene).

samples (tenfold diluted), the ORF1ab gene was totally not detected as opposed to N gene (S2 Table). Otherwise, nearly for all samples there is a linear correlation between the threshold reached and the doubling of the pool size (S3 Table).

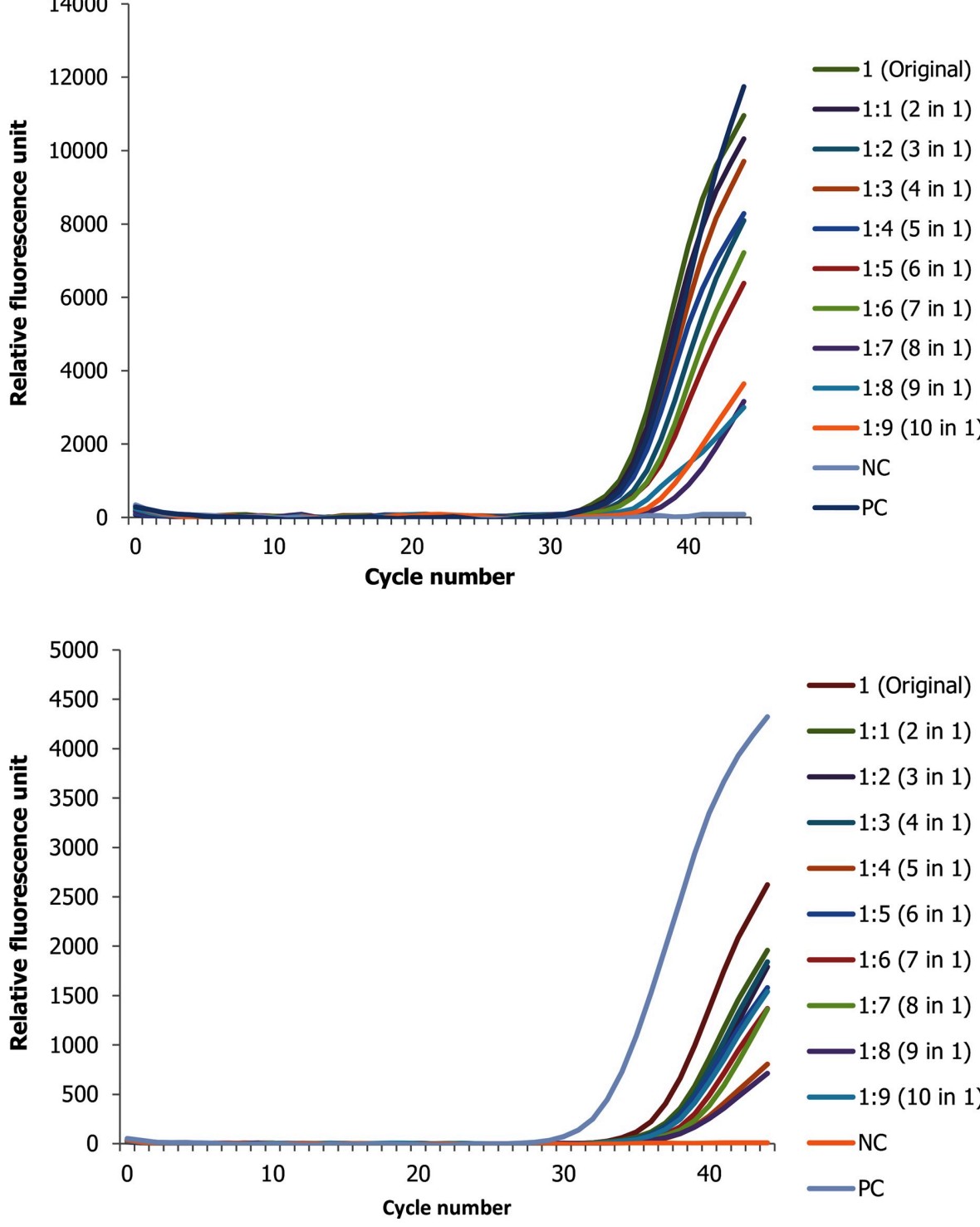

**Fig 3.** Change in ct value of positive RNA sample with high ct value (low viral copy) spiked with up to 9 negative samples for the two target genes (A N gene and B ORF1ab gene).

Using the online application http://www.chrisbilder.com/shiny, keeping all the numeric parameters indicated in the method section the same, we compared the influence of optimal pool size and disease prevalence rate on test efficiency. The data shows that when the disease

prevalence is between 1% to 7%, the optimal pool size ranges between 10 to 4, respectively. For a disease prevalence greater than 7%, the expected number tests reduced is less than 50% and the expected increase in test efficiency is less than 100% (S5 Table).

An important variable in the process of SARS CoV-2 testing that impacts epidemic control is the time interval between sample collection, sample delivery, testing, and result reporting. In our laboratory the potential maximum number of individual samples we can process over 24 to 36 hours is 276. However, using pooling of 4 samples in 1 we were able to process 1104 samples. This shows that our pooling strategy is robust and can significantly reduce turnaround time from 72–120 hours to 24–36 hours.

## Discussion

Globally, shortage of molecular laboratories for the diagnosis of SARS CoV-2, shortage of trained human capital, shortage of NA extraction, amplification and detection kits, and shortage of accessory and supplementary consumables despite an increasing number of testing demands has become an issue of concern [9, 10]. Particularly, the burden posed by these shortages is very high in Ethiopia given the pandemic has widely spread in a relatively short period of time. To minimize work load, resources and costs, a pooling approach for amplification and detection might be required. Here, we showed a proof-of-concept for direct clinical sample and RNA pooling for the diagnosis of SARS CoV-2 in Ethiopia using the existing assay.

Results from this pooling method supports that pooled screening strategy can be pursued to increase testing throughput, limit use of reagents, and to increase testing efficiency [7, 8], at an expected slight loss of sensitivity for direct clinical sample pooling. The same could be attained with no loss of sensitivity for RNA pooling. This study also showed that pooling is feasible using the current SARS CoV-2 assay in both public and clinical setting., implementing the method is especially imperative in resources limited the countries where the desire to test large number of individuals has been impacted by the shortage of key supply of detection kits. The predictive algorithm indicated a direct clinical sample pooling ratio of 5 in 1 was expected to retain accuracy of the test irrespective of the ct value of the sample spiked, and results in a 137% increase in test efficiency.

In general, the practical application of the pooling approach is confirmed in that it saved reagents, and reduced personnel time by three-fold that could expand testing. Assuming a consistent positivity rate in the country, pooling strategy on direct biological and RNA would expand testing by 168% and 120%, respectively. However, in a rapidly changing epidemic, testing strategies will need to adapt to real time situation. That is, a potential increases in the prevalence rate of a diseases requires the use of highly sensitive assays to avoid missing samples with low RNA copy number [8–10]. Furthermore, the impact of different extraction methods on the recovery of RNA/NA and overall assay sensitivity needs to be evaluated. And, thus both public and clinical laboratories must perform validation pool studies for their own methods of RNA/NA extraction and detection, to align their testing methods with the prevalence rates of SARS CoV-2 in real time of the settings. Because of the availability of limited SARS CoV-2 diagnosis facility, access to diagnostic tests, kit supplies, and the increasing number of individuals to be tested while there is shortage of trained human capital, this approach is important to facilitates rational use of resources. Furthermore, the approach could allow for prospective monitoring of the effectiveness of contact reduction measures at the population level and early detection of epidemic waves [11].

However, the limitation of this study is that because of the lack of a plasmid with known concentration, we were not able to quantify the changes occurred in between the dilutions in terms of viral copy number.

## Conclusion

Considering an increasing SARS CoV-2 epidemic and the possibility of unrecognized spread of the diseases within the community, we propose a rapid and straightforward screening strategy for SARS CoV-2 using either direct biological sample pooling of 4/5 in 1 or RNA pooling up to 8 in 1. We do not recommend pooling of clinical samples if the disease prevalence is greater than 7%; especially in case of large sample size. This approach proved its concept and principles, and may facilitate detection of early community transmission of SARS CoV-2 to enable the timely implementation of appropriate infection control measures to reduce spread. The method can also be used for routine monitoring of healthcare workers and individuals at higher risk of exposure.

## Supporting information

**S1 Table. Ct values of the original positive sample (with low Ct value highlighted in silver) and the pools from direct clinical samples, this corresponds to Fig 1A and 1B.**
(DOCX)

**S2 Table. Ct values of the original positive sample (with a high Ct value silver highlighted) and the pools from direct clinical samples.**
(DOCX)

**S3 Table. Ct values of the original RNA positive sample (with low Ct value highlighted in silver) and the RNA pools, this corresponds to Fig 2A and 2B.**
(DOCX)

**S4 Table. Ct values of the original RNA positive sample (with high Ct value highlighted in silver) and the RNA pools, this corresponds to Fig 3A and 3B.**
(DOCX)

**S5 Table. A comparison of the influence of optimal sample pool size and disease prevalence rate on test efficiency.**
(DOCX)

## Author Contributions

**Conceptualization:** Andargachew Mulu, Dawit Hailu Alemayehu, Alemseged Abdissa, Abebe Genetu Bayih, Getachew Tesfaye Beyene.

**Data curation:** Andargachew Mulu, Dawit Hailu Alemayehu, Fekadu Alemu, Dessalegn Abeje Tefera, Sinknesh Wolde, Gebeyehu Aseffa, Tamrayehu Seyoum, Meseret Habtamu, Getachew Tesfaye Beyene.

**Formal analysis:** Andargachew Mulu.

**Methodology:** Andargachew Mulu, Dawit Hailu Alemayehu, Fekadu Alemu, Dessalegn Abeje Tefera, Sinknesh Wolde, Gebeyehu Aseffa, Tamrayehu Seyoum, Meseret Habtamu, Alemseged Abdissa, Abebe Genetu Bayih, Getachew Tesfaye Beyene.

**Project administration:** Andargachew Mulu, Getachew Tesfaye Beyene.

**Supervision:** Andargachew Mulu, Meseret Habtamu, Alemseged Abdissa, Abebe Genetu Bayih, Getachew Tesfaye Beyene.

**Validation:** Sinknesh Wolde, Gebeyehu Aseffa, Tamrayehu Seyoum, Meseret Habtamu, Alemseged Abdissa, Abebe Genetu Bayih.

**Writing – original draft:** Andargachew Mulu, Dawit Hailu Alemayehu, Fekadu Alemu, Dessalegn Abeje Tefera, Meseret Habtamu, Alemseged Abdissa, Abebe Genetu Bayih, Getachew Tesfaye Beyene.

**Writing – review & editing:** Andargachew Mulu, Dawit Hailu Alemayehu, Fekadu Alemu, Dessalegn Abeje Tefera, Sinknesh Wolde, Gebeyehu Aseffa, Tamrayehu Seyoum, Meseret Habtamu, Abebe Genetu Bayih, Getachew Tesfaye Beyene.

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
