## [Decision Letter · Decision Letter 0]

7 Dec 2020

PONE-D-20-35695

Evaluation of Sample Pooling for Screening of SARS CoV-2

PLOS ONE

Dear Dr. Beyene,

Thank you for submitting your manuscript to PLOS ONE. After careful consideration, we feel that it has merit but does not fully meet PLOS ONE’s publication criteria as it currently stands. Therefore, we invite you to submit a revised version of the manuscript that addresses the points raised during the review process.

We look forward to receiving your revised manuscript.

Kind regards,

Etsuro Ito

Academic Editor

PLOS ONE

Additional Editor Comment:

The language should be improved.

Journal Requirements:

https://journals.plos.org/plosone/s/file?id=ba62/PLOSOne_formatting_sample_title_authors_affiliations.pdfú

2. Thank you for including your ethics statement:  "Being an Internal Experiment at methods validation level and given the decoded nature of testing, individual patient consent was not required and conducted based on institutional system.

And, the Armauer Hansen Research Institute/ALERT Ethics Review Committee has also waived it."

Please amend your current ethics statement to confirm that your named institutional review board or ethics committee specifically approved this study.

5. We noticed you have some minor occurrence of overlapping text with the following previous publication(s), which needs to be addressed:

- https://academic.oup.com/ajcp/article/153/6/715/5822023

- https://www.medrxiv.org/content/10.1101/2020.03.26.20039438v1

- https://jamanetwork.com/journals/jama/fullarticle/2764364

In your revision ensure you cite all your sources (including your own works), and quote or rephrase any duplicated text outside the methods section. Further consideration is dependent on these concerns being addressed.

Reviewers' comments:

Reviewer's Responses to Questions

**Comments to the Author**

1. Is the manuscript technically sound, and do the data support the conclusions?

Reviewer #1: Partly

Reviewer #2: Yes

2. Has the statistical analysis been performed appropriately and rigorously? 

Reviewer #1: N/A

Reviewer #2: Yes

3. Have the authors made all data underlying the findings in their manuscript fully available?

Reviewer #1: No

Reviewer #2: No

4. Is the manuscript presented in an intelligible fashion and written in standard English?

Reviewer #1: No

Reviewer #2: No

5. Review Comments to the Author

Reviewer #1: Mulu A and coworkers describe sample pooling for the screening of SARS-CoV-2 in Ethiopia. The authors performed the experiments both on direct clinical sample pooling and extracted RNA pooling.

Major comments:

1. One of the objectives of the study was to establish time and resource-saving method for the screening of SARS-CoV-2. But the authors did not show or discuss on the turnaround time of this strategy.

2. The RT-PCR kit describes in the Methods detected E and ORF1ab genes, but the results describe the Ct value of N and ORF1ab genes.

3. The Ct values of each pool and original sample should be demonstrated. The numbers of specimens and the numbers of pools should be shown.

4. The results described the ability of pooling strategy to detect the virus in the pool of direct clinical sample of up to eight samples. Why did the authors recommend the pooling of four samples? Also, in the extracted RNA, the pooling strategy detected the virus in the pool of 10 samples, but recommend the pool of eight. Please provide the reasons for the recommendation.

5. It might be more interesting if the authors describe that they performed the experiments in both direct clinical sample and extracted RNA in the abstract.

Minor comments:

1. COVID-19 stands for coronavirus disease 2019. The full name of the disease should be corrected both in the Abstract and Background.

2. The method for calculation of pool size should be moved to the Method section. Please specify the algorithm that the authors used in the website. Does the reduction in the expected number of tests were 741% for the pool of 10 samples in Table 2? Please describe the term “testing efficiency”.

3. The manuscript should be language check. There are many typos. Also, the writing can be improved by reorganizing the content.

Reviewer #2: The experimental study by Mulu et al proposed a simple and straightforward screening strategy for SARS CoV-2 using either direct clinical specimen or pooling of NA after the extraction step. Such a strategy has been demonstrated to be useful in the face of global scarcity of kits and consumables, among others, particularly in resource constrained settings. However, the manuscript requires some clarity in the methodology and findings. Specific comments to improve the manuscript include:

General:

Language edition needed, including revising long sentences

Method:

• Clearly indicate how many samples from each of Low, Medium, High Ct values categories were used in the study. Indicate their Ct values as well (for both target genes). In other words, how many different pools were made for each category of samples

• How were the low, medium and High Ct cutoffs determined?

• Lines 135 to 137: the sentence is not clear.

• When preparing the negative pool, indicate the volume of sample taken from each individual sample (just for clarity)

• Authors have stated tests were done in triplicate. It is not clear from the document that how the final result was determined and what decision were made if triplicates do not agree. Also state the degree of acceptable variation between triplicates

• For the reader, authors need to clarify how they determined rates of test efficiency in the method section (though shown as table footnote)

• Include a section on Quality Assurance

Result

• Complete the figures for the 3 cut off levels of positive samples pooled. In the result only the following figures were shown with no explanation why only those depicted (Fig 1=direct sample Low Ct, Fig 2=RNA Low Ct, Fig 3=RNA High Ct)

• Accordingly revise the discussion section

Discussion

• Lines 230231: Instead of using the phrase “ lack of” better to state “shortage of”

• Lines 230-234 needs revision; particularly the last phrase after the reference lacks continuity with the preceding statement (Lines 233-234)

• There is inconsistency in the suggested RNA pooling: Lines 249-250 states 10 in 1 while in the conclusion part it is stated 8 in 1

• Pooling strategy depends on the prevalence rate of SARS Cov-2. Can you suggest a rate that pooling is not recommended?

• With the increasing SARS Cov-2 prevalence, please comment on the applicability of the findings

Thank you!

6. PLOS authors have the option to publish the peer review history of their article (what does this mean?). If published, this will include your full peer review and any attached files.

Reviewer #1: No

Reviewer #2: No

---

## [Author Response · Author response to Decision Letter 0]

30 Jan 2021

Responses Reviewer

Reviewer #1: 

Mulu A and co-workers describe sample pooling for the screening of SARS-CoV-2 in Ethiopia. The authors performed the experiments both on direct clinical sample pooling and extracted RNA pooling.

Major comments:

1. One of the objectives of the study was to establish time and resource-saving method for the screening of SARS-CoV-2. But the authors did not show or discuss on the turnaround time of this strategy.

Response: Thanks a lot! this has now been included both in the result (last paragraph) and discussion section.

2. The RT-PCR kit describes in the Methods detected E and ORF1ab genes, but the results describe the Ct value of N and ORF1ab genes.

Response: It was my mistake that we wrote “E gene”, now corrected to N gene in all sections.

3. The Ct values of each pool and original sample should be demonstrated. The numbers of specimens and the numbers of pools should be shown.

Response: we included supplementary (S1-S4) Tables with the Ct value of the original positive samples and Ct values of the pools for both direct clinical samples and RNA samples 

4. The results described the ability of pooling strategy to detect the virus in the pool of direct clinical sample of up to eight samples. Why did the authors recommend the pooling of four samples? Also, in the extracted RNA, the pooling strategy detected the virus in the pool of 10 samples, but recommend the pool of eight. Please provide the reasons for the recommendation.

Response: Indeed, the pooling strategy using the specified NA extraction and RT-PCR protocol we can detect the virus in pool of 8 for direct clinical sample and in pool of 10 for RNA. However, considering the uncertainty in quality of samples, the efficiency of RNA extraction protocols, sensitivity of RT-PCR and possible human error in preparing pool of large samples, we recommended pool of 4/5 and 8. Furthermore, reports show that the gold standard test (detection of nucleic acids of SARS-CoV2 by RT-PCR) has up to 30% False negative reports. 

5. It might be more interesting if the authors describe that they performed the experiments in both direct clinical sample and extracted RNA in the abstract.

Response: Thanks again: We included pool of RNA in the abstract section too.

Minor comments:

1. COVID-19 stands for coronavirus disease 2019. The full name of the disease should be corrected both in the Abstract and Background.

Response: this is now corrected!

2. The method for calculation of pool size should be moved to the Method section. Please specify the algorithm that the authors used in the website. Does the reduction in the expected number of tests were 741% for the pool of 10 samples in Table 2? Please describe the term “testing efficiency”.

Response: Thank you very much, this question has helped us to re-analyse the calculation. We ask apologies that our calculation of the figures in the previous Table 2 were wrong. Now we have correctly calculated the most optimal testing configuration pool size, expected number of tests reduced, and testing efficiency using a Shiny App for pooled testing by Hierarchical algorithm. the previous Table 2 now is Table 3. By using disease prevalence rate 0.05, the calculated optimal pool size is 5 and the expected is number of tests reduced is 58%, with test efficiency of 137%.

3. The manuscript should be language check. There are many typos. Also, the writing can be improved by reorganizing the content.

Response: Thanks again, now the MS is read by native speaker. 

Thanks a lot for your comments and suggestions!

Reviewer #2: 

The experimental study by Mulu et al proposed a simple and straightforward screening strategy for SARS CoV-2 using either direct clinical specimen or pooling of NA after the extraction step. Such a strategy has been demonstrated to be useful in the face of global scarcity of kits and consumables, among others, particularly in resource constrained settings. However, the manuscript requires some clarity in the methodology and findings. Specific comments to improve the manuscript include:

General:

Language edition needed, including revising long sentences

Response: we revised and edited the Language 

Method:

• Clearly indicate how many samples from each of Low, Medium, High Ct values categories were used in the study. Indicate their Ct values as well (for both target genes). In other words, how many different pools were made for each category of samples

Response: Thank you, now we included in the method section that the total number of pools done on direct clinical sample were 54 using two positive and 18 negative samples. For the nucleic acid arm, a total of 18 pools were conducted using two positive and 16 negative samples. 

• How were the low, medium and High Ct cutoffs determined?

Response: This is really a very nice comment. As there could be subjectivity, and we could not come across any literatures that categorise Ct values in to Low, Medium and High, we just determined the cut-off values based on convince. Now we have modified the range of categories Ct values that is, we used the Ct value of the internal positive control as a reference. That is, Ct values less than or equal to the Ct value of the internal control (32) was considered as low ct value, and Ct values between >32 and < 34 were considered as medium and Ct value greater than 34 were considered as high. Please note that Ct value for negativity is 40 and above. 

• Lines 135 to 137: the sentence is not clear.

Response: We have revised this section 

• When preparing the negative pool, indicate the volume of sample taken from each individual sample (just for clarity)

Response: The sum volume of negative samples taken for dilution (pooling) indicated, for direct clinical samples in Table 1, and Table 2 for RNA. 

• Authors have stated tests were done in triplicate. It is not clear from the document that how the final result was determined and what decision were made if triplicates do not agree. Also state the degree of acceptable variation between triplicates

Response: We agree on the importance of showing the statistical variation of the experiments. We have used One-way ANOVA and analysed the average variance of pools Ct values between experiments and within, this is now indicated it result section. 

• For the reader, authors need to clarify how they determined rates of test efficiency in the method section (though shown as table footnote)

Response: Thanks, now we clearly stated in method section how test efficiency is calculated. We obtained test efficiency by dividing expected number of tests reduced by expected number of tests per individual. 

• Include a section on Quality Assurance

Response: We included quality assurance section in this revised version, at the end of method section. We also indicated that our laboratory has been participating in an external quality assessment program, and given the approval that results we produce are reliable.

Result

• Complete the figures for the 3 cut off levels of positive samples pooled. In the result only the following figures were shown with no explanation why only those depicted (Fig 1=direct sample Low Ct, Fig 2=RNA Low Ct, Fig 3=RNA High Ct)

• Accordingly revise the discussion section

Response: we have revised this part accordingly.

Discussion

• Lines 230-231: Instead of using the phrase “lack of” better to state “shortage of”

Response: we revised this section and we replace the “lack of” by “shortage of”

• Lines 230-234 needs revision; particularly the last phrase after the reference lacks continuity with the preceding statement (Lines 233-234)

Response: thank you, we revised this statement

• There is inconsistency in the suggested RNA pooling: Lines 249-250 states 10 in 1 while in the conclusion part it is stated 8 in 1

Response: thanks again, what we indicated in Lines 249-250 is the highest number of sample pool (9 negative and 1 positive samples) that we can get positive test. However, in the conclusion part, considering the uncertainties in the quality of samples, the efficiency of RNA extraction protocols, sensitivity of RT-PCR, and the possible human error in preparing pools of large samples, we recommended pooling of 8 samples in 1 for RNA.

• Pooling strategy depends on the prevalence rate of SARS Cov-2. Can you suggest a rate that pooling is not recommended?

Response: Although we did not see the impacts of pooling in community-based samples, we suggest using pooling in a community samples with estimated prevalence of less than or equal to 7%. However, we don’t advise pooling of clinical samples when disease prevalence is greater than 7%; particularly when sample size is large. Because, a disease prevalence of greater than 7% has a benefit of less than 50% in terms of the expected number of tests reduced.

• With the increasing SARS Cov-2 prevalence, please comment on the applicability of the findings. 

Response: with the increasing SARS Cov-2 prevalence the benefit of pooling is very minimal especially, when processing a very large volume sample. For example, suppose that the disease prevalence is 8%, if you are processing 1000 samples in a pooling of 4 samples in 1, you will have 250 pools. Out the 250 pools, potentially 80 of your pools will have positive signal. Then, in the subsequent step, you have to process 4x80 = 320 samples separately to find out which of the sample/s from each pool is truly positive. From this we can conclude that using sample pooling in such situation is a waste of time and resource.

Thank you very much for your comments and suggestions!

---

## [Decision Letter · Decision Letter 1]

15 Feb 2021

Evaluation of Sample Pooling for Screening of SARS CoV-2

PONE-D-20-35695R1

Dear Dr. Beyene,

We’re pleased to inform you that your manuscript has been judged scientifically suitable for publication and will be formally accepted for publication once it meets all outstanding technical requirements.

Kind regards,

Etsuro Ito

Academic Editor

PLOS ONE

Reviewers' comments:

Reviewer's Responses to Questions

**Comments to the Author**

1. If the authors have adequately addressed your comments raised in a previous round of review and you feel that this manuscript is now acceptable for publication, you may indicate that here to bypass the “Comments to the Author” section, enter your conflict of interest statement in the “Confidential to Editor” section, and submit your "Accept" recommendation.

Reviewer #1: All comments have been addressed

Reviewer #2: All comments have been addressed

2. Is the manuscript technically sound, and do the data support the conclusions?

Reviewer #1: Yes

Reviewer #2: Yes

3. Has the statistical analysis been performed appropriately and rigorously? 

Reviewer #1: Yes

Reviewer #2: Yes

4. Have the authors made all data underlying the findings in their manuscript fully available?

Reviewer #1: Yes

Reviewer #2: Yes

5. Is the manuscript presented in an intelligible fashion and written in standard English?

Reviewer #1: Yes

Reviewer #2: Yes

6. Review Comments to the Author

Reviewer #1: (No Response)

Reviewer #2: (No Response)

7. PLOS authors have the option to publish the peer review history of their article (what does this mean?). If published, this will include your full peer review and any attached files.

Reviewer #1: No

Reviewer #2: No

---

## [Editor Report · Acceptance letter]

17 Feb 2021

PONE-D-20-35695R1 

Evaluation of sample pooling for screening of SARS CoV-2 

Dear Dr. Beyene:

I'm pleased to inform you that your manuscript has been deemed suitable for publication in PLOS ONE. Congratulations! Your manuscript is now with our production department. 

Kind regards, 

on behalf of

Prof. Etsuro Ito 

Academic Editor

PLOS ONE